# Research on the Influence Mechanism of Fashion Brands' Crossover Alliance on Consumers' Online Brand Engagement: The Mediating Effect of Hedonic Perception and Novelty Perception

**Jinjiang Cai [1], Jingjing Wu [2], Hongjie Zhang [1,\*] and Yifei Cai [1]**

[1]   College of Textiles and Apparel, Quanzhou Normal University, Quanzhou 362000, China
[2]   College of Management and Economics, Fujian Agriculture and Forestry University, Fuzhou 350002, China
\*   Correspondence: zhjie016@qztc.edu.cn

**Abstract:** In recent efforts, instead of the conventional co-branding marketing approach, many fashion brands have tried to break through the original image by applying a crossover alliance method, and have achieved good results in practice. However, whether this kind of alliance can effectively enhance consumers' online brand engagement is still a key question to be addressed. Using the S-O-R model, the paper introduces two mediating variables, novelty perception and hedonic perception, to explore the potential mechanism of brand image differences and product type differences on consumers' online brand engagement under the background of a crossover alliance of fashion brands. This study shows that (1) brand image differences and product type differences positively affect consumers' novelty perceptions and hedonic perceptions, and (2) such positive perceptions facilitate online brand engagement of consumers.

**Keywords:** fashion brands; crossover alliance; online brand engagement; hedonic perception; novelty perception





## 1. Introduction

The rapid development of digital technology has made social media popular these days. Various social media sites have penetrated and become an integral part of consumers' daily lives, while companies are increasingly focusing on and valuing the ability of social media to contribute to their businesses [1]. More and more enterprises are using social media to post content related to their products in order to generate consumer excitement and increase brand engagement (likes, shares, comments, etc.) in the business competition [2]. Especially for products with high attention and low engagement such as apparel, when many fashion brands try to take a crossover alliance approach to launch new products, it is necessary to attract consumers' attention and motivate them to participate in brand-related activities online to have more awareness and influence. For example, Dior and Air Jordan 1 High OG collaborated on a co-branded sneaker that attracted great attention in social media and fetched a high price of tens of thousands of dollars on second-hand platforms; Gucci and The North Face crossover alliance, once the preview has caused a buzz in social networks, and Balenciaga's crossover collaboration with Fortnite, the online game, which simultaneously launched online game virtual sets and offline physical clothing, have achieved excellent business performance. The above cases show that the crossover alliances of fashion brands have gained good market feedback, both in the offline consumer market and the online stream. However, it is worth noting that there are also some cases of crossover alliances that have had the opposite effect. For example, the joint H&M and Kenzo model was hailed by netizens as "the most difficult to wear collection ever". The ZX 7000 sneakers jointly launched by HEYTEA and Adidas Originals were also

criticized on social media. Consequently, how enterprises can develop effective crossover alliance strategies to help fashion brands gain market share and enhance consumer brand engagement on social media is a critical issue that needs to be addressed.

In the apparel and fashion industry, brand association strategy has been one of the preferred ways for companies to explore new market values and gather a customer base. Brand association is a marketing strategy in which two or more similar brands cooperate to create new products [3]. The associations of fashion brands combine the respective characteristics of the constituent brands and transfer the relevant values into the co-branded products [4]. The aim is to drive discovery and familiarity with branded products among consumers who are not aware of the brand. Previous studies on the topic of brand alliances have given extensive attention to the fast-moving consumer goods and electronics markets [5], while only some studies have focused on fashion brand companies, analyzing fashion brand business model innovation [6], consumer brand loyalty [7], and the impact of fashion brand alliances on brand equity [4,8]. Scholars have emphasized the key factors of successful brand association based on the concept of "brand association similarity" [9], and have shown that similarity in brand image, product category, product attributes, and product quality among the collaborating brands can trigger consumer perception of fit [10–15], and a higher perception of fit has a positive impact on brand evaluation [3]. However, mechanisms based on co-branding similarity and perceived fit do not explain the existing successful cases of brand crossover cooperation in fashion marketing well.

It is well known that the fashion industry has always been a competitive and fast-changing market, and fashion brands need to continuously push the boundaries to ensure market growth, enhance consumer recognition, and ultimately maintain a competitive advantage [16], but which model of fashion brands' crossover alliance influences consumer perception and engagement remains unclear and requires further research. In addition, research has shown that consumer online brand engagement is an important strategic factor [17]. This is because companies channel offline activities and online streams into cyber environments, where they can track consumers' digital footprints more accurately and further delve into customer preferences [18]. However, existing research on fashion brand alliances only focuses on the effects on consumers' behavioral decisions such as purchase intention [19,20], payment premium [21], and willingness to recommend to others [22]. In the meantime, the understanding of consumers' online brand engagement remains limited, and little exploration has been conducted on the relationship between fashion brand crossover alliances and consumers' online brand engagement.

To settle the above issues, this study has two objectives: first, this study divides fashion brand crossover alliances into brand level and product level—in other words, brand image difference and product type difference—and argues that brand alliances with low fit can also have positive impacts, thus enriching the research related to brand extensions. The second objective is to clarify how fashion brand crossover alliances affect consumers' online brand engagement. Specifically, using the Stimulus–Organism–Response (S-O-R) theory, two mediating variables, consumer novelty perception and hedonic perception, are introduced to explore the potential mechanisms by which brand image differences and product type differences in fashion brand crossover alliances affect the formation of consumer online brand engagement. In terms of industry characteristics, consumer brand engagement on social media is particularly important for fashion brands [23,24]. Although consumer online brand engagement may contribute to the formation of relationships such as familiarity, loyalty, and trust [18], there is less research in the existing literature on the relationship between crossover alliances of fashion brands and consumer online brand engagement. This study enriches the antecedent theory of consumer online brand engagement and also provides a reference for companies to carry out fashion brand crossover alliances to enhance social media engagement.

## 2. Theoretical Backgrounds

### 2.1. Crossover Alliance of Fashion Brands

Brand association originates from the concept of symbiotic marketing [9], which refers to an effective way for two or more brands to form a short-term or long-term collaboration with the aim of creating synergies and complementing each other's resources to broaden the market and increase the brand value [3,25–27]. The number of business practices and academic research literature on brand alliances has increased significantly over the past few years, as companies have become increasingly aware of the growth opportunities offered by brand alliance strategies [28]. Previous studies have focused on exploring the impact of factors such as consumers' brand loyalty, perceived brand quality, and brand fit on brand alliance success [5,29]. It is emphasized in the study that among brand association partners, either brand image fit, product category fit, or consumer perception fit affects consumer perception fluency and further influences brand association attitudes [8,30]. However, with the evolution of the business market, high fit brand alliances in the field of fashion marketing are no longer sufficient to meet market and consumer needs. For instance, Moon and Sprott (2016) [31] showed that luxury brands may find it almost impossible to collaborate with other luxury brands and therefore need to consider non-luxury brand partners. Co-partners with too similar brand images or product types may struggle to inspire freshness in consumers and thus reduce purchase intentions [32]. Furthermore, it has been emphasized that moderate dissonance has a positive and interesting value. Studies by Srivastava and Sharma (2012) [33] and Walchli (2007) [34] pointed out that in the brand extension process, the degree of brand extension consistency shows an inverted U-shaped relationship with brand extension evaluation, i.e., moderate incongruence leads to better brand evaluation than homogeneity or extreme incongruence. Based on product innovation domain theory, Zhu and Wang (2016) [35] emphasized the importance of consumer novelty perceptions in the brand innovation process. Consumers may have stronger novelty perceptions and hedonic perceptions for innovative products that deviate from the norms. These phenomena may be due to the possibility that consumers find the brand or product more unique, iconic, or stylish [36,37].

As a result, from luxury brands to fast fashion brands, all companies try to break the original brand image, seek differentiation, and create a "brand crossover alliance" [38]. A brand crossover alliance is a form of brand alliance, which is defined by scholars as the joint development of new products by two or more brands in non-competing industries that highly integrate the characteristics or attributes of the joint brand [39]. Furthermore, some scholars argue that "crossover" cannot be simply summarized as the difference between industries or sectors in an objective sense. The so-called crossover should be a brand association partner that consumers perceive to be less related, similar, compatible, or complementary in terms of their salient attributes [40]. In this way, a crossover alliance between fashion brands can be a collaboration between two brands with widely different images in the fashion industry. For example, Gucci and The North Face are both fashion brands, but in terms of brand image, one is a luxury brand and the other is an outdoor sports brand. The perceived similarity of their brand images is rather small, and the product positioning or product values are very different. Oeppen and Jamal (2014) [41] and Mrad et al. (2019) [42] explored through qualitative research the good results achieved by fast fashion brands and luxury brands through low-scale collaborations. This is because it seems that through short-term collaborations, luxury brands can be protected from being viewed negatively by their existing consumer base while also attracting potential consumers to new markets through the mass market of fast fashion brands. On the other hand, the crossover alliance of fashion brands can also be the collaboration of the fashion industry with other industries. For example, Balenciaga and Moncler successively designed virtual clothing with the online game Fortnite. In terms of product type, one is an Internet game company and the other is a luxury fashion brand, and both product types have strong differences in terms of product attributes and product usage scenarios. The case study by Moon and Sprott (2016) [31] showed that when fashion brands and brands from



other sectors unite, consumers' perceived brand image fit positively influences purchase intentions for co-branded new products. The study by Paydas (2021) [21] showed that brand alliances of different product types positively influence consumers' perceptions of product quality and enhance willingness to purchase the product.

Reviewing the above literature, we find that scholars analyze the effectiveness of fashion brand crossover alliances from the perspective of brand image difference or product type difference, respectively, but fewer scholars incorporate the two dimensions into one model for exploration and comparison. Thus, this study explores fashion brand crossover alliances from two dimensions: brand image difference and product type difference, and defines fashion brand crossover alliances as the joint development of new products by a fashion brand with single or multiple brands that have outstanding differences at the brand image level or product type level. Differences at the brand image level refer to consumers' perceptions that the brand image, brand positioning, and brand perception of the partners are not similar; differences at the product type level refer to consumers' perceptions that the product attributes and product usage contexts of the partners are not compatible and complementary.

### 2.2. Online Brand Engagement of Consumers

The concept of consumer online brand engagement is derived from the service-dominant logic of Relationship Marketing Theory [43], but there is still no consensus among academicians on the definition of the concept. Engagement is a multifaceted concept that encompasses a range of cognitive, affective, and behavioral dimensions [17]. Brand-related online behaviors such as browsing, consuming, commenting, and sharing can be conceptualized as online brand engagement [44]. Muntinga et al. (2021) [45] dissected three structures of consumers' online brand engagement in conjunction with the COBRAs framework. One is "consumption", where consumers may only engage but not post content, such as watching videos or reading reviews. This category is also referred to as "passive users" [46]. The second is "contribution", where consumers may join brand communities to communicate and discuss, or comment on relevant content posted by brands. The third is "creation", in which consumers create brand-related content that is viewed and discussed by other participants of the brand. For example, consumers are transformed into active brand participants [17] and creators of brand stories [47] through social networks. Similarly, Khan (2017) [46] divided consumer online brand engagement behavior into two structures, namely consumption and engagement. Consumption includes consumers watching and reading branded content, while engagement includes consumer likes, feedback, recommendations, and conversations. Since this study aims to understand consumers' reactions to new products combined by brands across borders, it focuses on consumer engagement, defining consumer online brand engagement as consumers expressing likes, sharing comments, and recommending purchases of apparel (or products) combined by fashion brands across borders on social media.

The large user base of social media (e.g., Facebook, Twitter, Instagram, and LinkedIn) has brought its importance to the attention of more brands [48]. The interactive, participatory, and open nature of social media provides an easier and more effective channel for brands to discover consumer needs and build connections [49]. As the market becomes more competitive, brands want to build close connections with consumers through various online marketing promotions, rather than simply a single purchase relationship [50,51]. Therefore, seeking consumer online brand engagement has become of growing importance to companies. The existing literature suggests that either intrinsic consumer-based perceptions or external cue-based factors are key antecedents influencing consumer online brand engagement. Based on intrinsic consumer perceptions, factors such as consumer satisfaction, trust, brand commitment, brand attachment, and perceived brand performance are important factors influencing consumer engagement [52]. However, most of them are explored based on traditional brand marketing scenarios, and there may be different key drivers such as novelty perception and hedonic perception in brand crossover alliance

scenarios. Studies by Zhu et al. (2016) and Dun et al. (2020) [35,53] both emphasized the importance of novelty perception in the brand product innovation process, which has a significant impact on consumers' purchase intentions. In addition, Kim et al. (2023) [24] showed that hedonic perception has a significant role between brand advertising interactivity and consumers' purchase intention. Despite the fact that scholars have explored the influence of novelty perception and hedonic perception on consumers' purchase decisions, there is a need to further understand the impact of novelty perception and hedonic perception on consumers' online brand engagement in the context of crossover alliances from the perspective of fashion companies. Based on the external cues aspect, consumers are more likely to engage with social media content that introduces new products with low consistency [54]. In line with this, Borah et al. (2020) [55] suggest that topical, timely, or unexpected improvised content items can facilitate consumer conversations with brands on social media. Lee et al. (2018) [56] state that content related to brand personality (e.g., humor and emotion) inspires higher levels of consumer engagement. Thus, product content that differs from traditional brand associations, such as new product content from fashion brands associated across borders, may be appropriate for the social media environment, but the impact of brand crossover alliances on consumer online brand engagement and its underlying mechanisms need to be further investigated.

### 2.3. Stimulus–Organism–Response (S-O-R) Theory

Mehrabian and Russell (1974) introduced the "organism" factor from the perspective of environmental psychology. They studied the internal response of individuals to environmental stimuli and proposed the "Stimulus-Organism-Response" theory. The S-O-R model states that individuals' behavioral decisions are not always rational and that they may be influenced by external stimuli (e.g., products or situations) (S) that affect their internal perceptions, cognition, or subconscious (O) and drive behavioral responses (R) [57]. The use of the S-O-R model allows a better connection between external stimuli and consumers' internal perceptions and reactions [36] and has been widely used in online social media scenarios to analyze consumer responses and behavioral preferences [24,58]. For example, Casaló and Ibáñez-Sánchez (2021) [58] found that the content (S) posted by fashion brands in social media influences consumers' perceived creativity and positive emotions (O) toward the brand, which increases emotional commitment and willingness to interact (R) with the brand. Kim et al. (2023) [24] found that the interactivity (S) of fashion brand advertising triggers consumers' hedonic perceptions (O) and increases consumers' purchase (R).

While previous studies have used branded online advertising as a stimulus, they have not examined how specific features of brand advertising (e.g., the particular product message of fashion brand crossover alliances) impact consumer perceptions and behaviors. Therefore, this study extends the S-O-R framework to the domain of fashion brand crossover alliances research. For a peripheral stimulus cue (i.e., fashion brand crossover association), the S-O-R framework is used to explore the impact of fashion brand crossover alliances on consumers' online brand engagement. The brand image difference and product type difference of a fashion brand crossover alliance are used as external environment stimulus variables (S), hedonic perception and novelty perception as individual cognitive psychological variables (O), and consumer online brand engagement as the driving behavioral response (R). Shown in Figure 1.

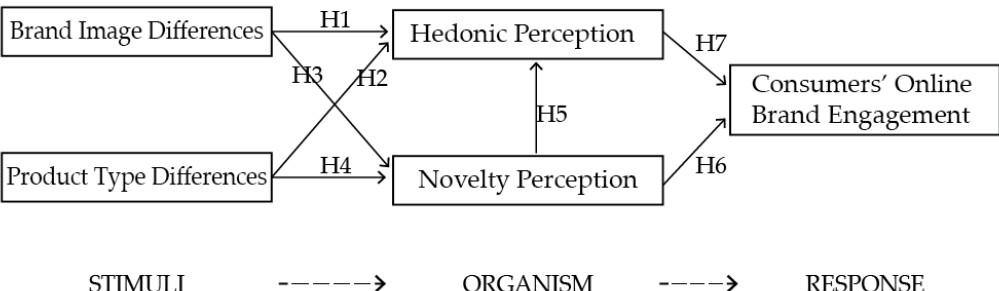

**Figure 1.** Conceptual model.

## 3. Research Hypotheses

### 3.1. Brand Image Differences, Product Type Differences, and Hedonic Perceptions among Alliance Partners

A crossover alliance of brands allows collaborating brands to share their values, as well as to have higher resource integration and better market feedback [59]. Concerning brand image, the crossover alliance of two well-known and distant fashion brands gives the advantage of the brand value linkage effect. Specifically, fashion products belong to products that highlight self-expression and favor hedonism, which are carriers for consumers to express their self-identity and consumer status [60]. Rajput et al. (2012) [61] stated that consumers wear fashionable clothes to show their distinctiveness and self-image. Khallouli and Gharbz (2013) [62] stated that young people consume fashion not for the product itself but for the symbolic meaning of a particular garment, such as self-expression, social image, etc. As a consequence, the mass prestige of the brand helps them to gain hedonic perceptions [63]. In terms of product type, two brand federations with widely different product types of crossover alliance break consumers' previously fixed perceptions of the product, making it more symbolic of exclusivity, being unique-making, or scarcity. Consumers' desire for uniqueness is related to the hedonic benefits of the product [45]. Khan (2017) [46] argued that consumers who pursue hedonic goals are more focused on product specificity and uniqueness than consumers who do not pursue hedonic goals. De Vries and Carlson (2014) [47] argued that the atypical design of products has a positive effect on consumers' hedonic perceptions because atypical design products are more likely to awaken consumers' interest. Based on the above analysis, this study proposes the following hypotheses.

**Hypothesis 1 (H1):** *Brand image differences have a positive impact on consumers' hedonic perceptions in the crossover alliance of fashion brands.*

**Hypothesis 2 (H2):** *Product type differences have a positive impact on consumers' hedonic perceptions in the crossover alliance of fashion brands.*

### 3.2. Brand Image Differences, Product Type Differences, and Novelty Perceptions among Co-Brandings

Seeking novelty has been the fashion industry's way to gaining a larger consumer market. In terms of brand image, the emergence of the co-branding model breaks the stereotypes consumers have about the original brand, leading to more associations with the brand personality, and bringing freshness and interest [39]. Sénéchal et al. (2014) [32] argued that collaboration between similar brands may not be perceived as innovative and that incompatible brand association strategies are more likely to be perceived as original, leading to high novelty perceptions. Shan et al. (2022) [64] found that when product category fit is high, low brand image and brand personality fit (e.g., fast fashion brands H&M and luxury brands Balmain unite) results in novelty perception, which leads to a positive evaluation of the co-brand. In terms of product type, product novelty is generally expressed in terms of external characteristics and expressions (e.g., product shape, color, or material) [65]. Goode et al. (2013) [66] argued that product novelty perception is a subjective

judgment of consumers comparing whether a product is more unique, different, etc., than similar products. A moderate degree of incongruity is more evocative and stimulating in nature, resulting in a greater sense of novelty [33]. The study of Jian et al. (2021) [40] also showed that crossover allied products cause consumers to have a good experience of surprise, surprising yet reasonable, and thus inspiring. Therefore, based on the above analysis, this paper proposes the following hypothesis:

**Hypothesis 3 (H3):** *Brand image differences have a positive impact on consumers' novelty perception in the crossover alliance of fashion brands.*

**Hypothesis 4 (H4):** *Product type differences have a positive impact on consumers' novelty perception in the crossover alliance of fashion brands.*

### 3.3. Novelty Perception and Hedonic Perception

Several studies have demonstrated that novelty has a significant impact on consumer hedonic perceptions. Moon et al. (2018) [67] showed that novelty-seeking perceptions mediate between novelty and hedonic value, with novelty influencing coolness, which in turn influences hedonic value, which in turn influences consumer attitudes. Research by Zaggl et al. (2019) [68] testing the mass customization of sneakers suggested that consumer-generated novelty perceptions are conditional on perceived hedonic benefits at the attribute level, and that differences and innovative solutions are more likely to be found in hedonic product attributes. Alba and Williams' (2013) [69] study revealed the association between the novelty of product design and consumer pleasure. Based on the above analysis, this study proposes the following hypothesis.

**Hypothesis 5 (H5):** *Novelty perception has a positive impact on consumers' hedonic perception in the crossover alliance of fashion brands.*

### 3.4. Novelty Perception and Consumers' Online Brand Engagement

Aiming to create a willingness to engage with brands, on social media platforms, brands strive to stand out and attract the attention of consumers [55]. Novel content evokes more unique information cues and external stimuli, as individuals are stimulated by novelty in their environment to generate unfamiliar perceptions, which stimulate interest in the content. As interest increases, it evokes the desire of the individual to go for further and more relevant information [70]. Tafesse (2015) [71] showed that consumer novelty perception becomes an important driver of brand-post liking. On the Facebook pages of automotive brands, as the novelty of brand posts increases, the likelihood of viewers clicking and commenting increases. Eelen et al. (2015) [72] showed that simply adding the "new" tag to a product helps increase consumer curiosity and promotes brand engagement intentions. Gerrath et al. (2021) [54] concluded that less consistent new products have a positive impact on brand engagement due to the fact that consumers are more curious about less consistent products, which drives them to engage more with the brand's social media. Based on the above analysis, this study proposes the following hypothesis.

**Hypothesis 6 (H6):** *Novelty perception has a positive impact on consumers' online brand engagement in the crossover alliance of fashion brands.*

### 3.5. Hedonic Perception and Consumers' Online Brand Engagement

In accordance with Kevin et al.'s (2017) [73] research has shown that one of the four factors influencing the engagement of luxury brands on Facebook and Instagram is the "ideal social self-image", which is part of the self-concept and relates to how the consumer wants others to perceive him or her. In addition, the hedonic attributes of a product are often associated with positive feelings, such as the ability to provide a sense of superiority [74]. Some consumers use and engage with brands in an attempt to send signals to others

about their relevance [75]. As also pointed out by Wallace et al. (2014) [76], consumers connect with brands to interact with others on the brand network rather than to establish a connection with the brand. For this reason, when the crossover alliance of brand image differences stimulates consumers' hedonic perceptions, they may be more likely to go to social media to interact with them because the consumer experience is reflective of their social identity. For illustration, the prestige associated with a brand induces consumers to build a relationship with the brand online [63]. A consumer may want a friend to know if he/she purchased an Armani shirt, but if they purchased a cheaper shirt at Walmart, they may not spread and share it widely [77]. Stathopoulou et al. (2017) [78] expressed that creative advertising may drive engagement on social media. Unique and scarce products have higher conversational value, which means that consumers can initiate interpersonal interactions by word-of-mouth sharing about the product. Therefore, when the crossover alliance of product type differences stimulates consumers' hedonic perceptions, they may be more willing to share unusual product content. Based on the above analysis, this study proposes the following hypothesis.

**Hypothesis 7 (H7):** *Hedonic perception has a positive impact on consumers' online brand engagement in the crossover alliance of fashion brands.*

## 4. Methodology and Study Design

### 4.1. Survey Methodology and Sample Selection

The questionnaire method was used in this study. The questionnaire contains basic information such as the main part and demographics, such as gender, age, education, etc. The main part includes an investigation of five latent variables: brand image difference (3 questions), product category difference (3 questions), hedonic perception (4 questions), novelty perception (4 questions), and brand engagement in social media (5 questions), with the questions for each variable adapted from the literature of related studies (the specific scale is shown in Table 1), and the Likert five-point scale method (1 = completely disagree; 5 = completely agree) to classify the options.

This study focuses on the impact of fashion brands' crossover alliances on consumers' online brand engagement. First, six groups of fashion brands' crossover alliance cases that are better known and more influential in the market were selected, including cases of different brand images (Gucci and The North Face, Dior and Air Jordan 1 High OG, and LOUIS VUITTON and Supreme) and different product types (HEYTEA and Adidas Originals, Balenciaga x and Fortnite, and LOUIS VUITTON and BMW). The questionnaire began by explaining the concept of the fashion brands' crossover alliance to respondents and showing consumers one of the six groups of fashion brands' crossover alliance cases. This step was taken to ensure that all respondents had an understanding of the research questions. A screening question was set (Are you aware of fashion brands' crossover alliances? Yes/No) to ensure the applicability of the respondents to the purpose of the study. Respondents answered "yes" to be included in the valid sample. The questionnaire was distributed and collected online using the "WJX (platform providing functions equivalent to Amazon Mechanical Turk)" from August 1 to 16, 2022. A total of 600 questionnaires (100 per case) were returned. After eliminating 21 non-conforming questions and 53 questionnaires with incomplete responses, a valid sample of 526 people was finally obtained.

As shown in Table 2, 74% of the respondents in the study sample were between the ages of 18 and 39, and 79% of the respondents had a monthly income of CNY 5000 or more. It can be seen that most of the respondents under the survey are young and middle-aged, who are more willing to accept new business formats and are the main consumers of fashion brands' crossover alliance business formats. The respondents are therefore in line with the target group of this study.

**Table 1.** Survey Scale.

| Latent Variable | Index | Question | Source |
|---|---|---|---|
| Product type differences (PD) | PD1 | Clothing (or products) of fashion brands that crossover and combine with company A and company B differ in terms of product characteristics. | [79] |
| | PD2 | Clothing (or products) of fashion brands that crossover and combine with company A and company B differ in terms of usage scenarios. | |
| | PD3 | Clothing (or products) of fashion brands that crossover and combine with company A and company B differ in terms of the needs they meet. | |
| Brand image differences (BD) | BD1 | Fashion brands cross over to Company A and Company B differ in terms of brand image. | [80] |
| | BD2 | Fashion brands cross over to Company A and Company B differ in terms of brand positioning. | |
| | BD3 | Fashion brands cross over to Company A and Company B differ in terms of brand perception. | |
| Hedonic perception (HP) | HP1 | Fashion brands' crossover allied clothing (or products) bring me a lot of new experiences. | [81] |
| | HP2 | Fashion brands' crossover allied clothing (or products) bring me a lot of fun. | |
| | HP3 | Fashion brands' crossover allied clothing (or products) bring me a lot of satisfaction. | |
| | HP4 | I like the clothes (or products) result from fashion brands' crossover alliance. | |
| Novelty perception (NP) | NP1 | Fashion brands' crossover allied clothing (or products) is different. | [82] |
| | NP2 | Fashion brands' crossover allied clothing (or products) is original. | |
| | NP3 | Fashion brands' crossover allied clothing (or products) is attractive. | |
| | NP4 | Fashion brands' crossover allied clothing (or products) is attractive. | |
| Online brand engagement (OE) | OE1 | I will comment on the clothing (or product) of fashion brands' crossover alliance on social media. | [83] |
| | OE2 | I will repost the clothing (or product) of the fashion brands' crossover alliance on social media. | |
| | OE3 | I will share fashion brands' crossover allied clothing (or products) with my friends on social media. | |
| | OE4 | I will buy fashion brands' crossover allied clothing (or products) and share them on social media. | |
| | OE5 | I will look forward to seeing more fashion brands crossover allied clothing (or products) on social media. | |

**Table 2.** Sample characteristics.

| Variables | Definition | Frequency (N = 526) | Percentage |
|---|---|---|---|
| Gender | Male | 257 | 49% |
| | Female | 269 | 51% |
| Age | Under18 | 16 | 3% |
| | 18–29 | 179 | 34% |
| | 30–39 | 210 | 40% |
| | 40–49 | 73 | 14% |
| | Above 51 | 48 | 9% |
| Education | Middle school | 10 | 2% |
| | High school | 17 | 3% |
| | junior college | 68 | 13% |
| | Bachelor | 314 | 60% |
| | Master degree or more | 117 | 22% |
| Average monthly income | CNY < 2500 | 19 | 4% |
| | CNY 2501–3000 | 37 | 7% |
| | CNY 3001–5000 | 55 | 10% |
| | CNY 5001–8000 | 252 | 48% |
| | CNY > 8000 | 163 | 31% |

### 4.2. Reliability and Validity Test of Samples

Before testing the hypotheses, validated factor analysis (CFA) was performed on all latent variables using AMOS 24.0 to assess the reliability and validity of the measurement model. First, the factor measurement model had a good fit ($\chi^2$ = 241.467; $\chi^2$/df = 1.700; $p$ = 0.000; GFI = 0.955; AGFI = 0.940; CFI = 0.991; TLI = 0.989; RMSEA = 0.037), and therefore, a reliability test could be performed. Secondly, the results are shown in Table 3, where the standardized factor loadings (Std.) of all five constructs exceeded 0.70 ($p$ < 0.001), the squared multiple correlations (SMC) exceeded 0.50, and the combined reliability (CR) exceeded 0.70, which met the evaluation criteria of Forza (2002) [84], indicating that the measurement model had good reliability. Second, the validity of the measurement model was judged by convergent validity and discriminant validity. According to the evaluation criteria of [68], the AVE values of all latent variables exceeded 0.50, which indicated that the measurement model had good convergent validity. Furthermore, as shown in Table 4, the square root of the AVE of each latent variable is greater than the Pearson correlation coefficient between variables, which meets the evaluation criteria of Podsakoff et al. (2012) [85], indicating that the measurement model has good discriminant validity.

**Table 3.** CFA Results.

| Latent Variable | Item | Parameter Significance Estimation | | | | Item Reliability | | Composite Reliability | Convergent Validity |
|---|---|---|---|---|---|---|---|---|---|
| | | Unstd. | S.E. | z-Value | P | Std. | SMC | CR | AVE |
| Product type differences | PD1 | 1.000 | | | | 0.983 | 0.966 | 0.927 | 0.809 |
| | PD2 | 0.948 | 0.029 | 33.080 | *** | 0.879 | 0.773 | | |
| | PD3 | 0.916 | 0.032 | 28.853 | *** | 0.830 | 0.689 | | |
| Brand image differences | BD1 | 1.000 | | | | 0.853 | 0.728 | 0.906 | 0.763 |
| | BD2 | 1.005 | 0.040 | 25.038 | *** | 0.886 | 0.785 | | |
| | BD3 | 0.993 | 0.040 | 24.915 | *** | 0.881 | 0.776 | | |
| Hedonic perception | HP1 | 1.000 | | | | 0.879 | 0.773 | 0.921 | 0.745 |
| | HP2 | 1.021 | 0.037 | 27.763 | *** | 0.886 | 0.785 | | |
| | HP3 | 0.981 | 0.040 | 24.272 | *** | 0.821 | 0.674 | | |
| | HP4 | 0.968 | 0.036 | 26.677 | *** | 0.866 | 0.750 | | |
| Novelty perception | NP1 | 1.000 | | | | 0.872 | 0.760 | 0.929 | 0.765 |
| | NP2 | 0.983 | 0.037 | 26.364 | *** | 0.861 | 0.741 | | |
| | NP3 | 1.000 | 0.035 | 28.829 | *** | 0.905 | 0.819 | | |
| | NP4 | 0.992 | 0.038 | 26.238 | *** | 0.859 | 0.738 | | |
| Online brand engagement | OE1 | 1.000 | | | | 0.979 | 0.958 | 0.959 | 0.824 |
| | OE2 | 0.968 | 0.023 | 41.988 | *** | 0.901 | 0.812 | | |
| | OE3 | 0.937 | 0.024 | 38.975 | *** | 0.884 | 0.781 | | |
| | OE4 | 0.960 | 0.024 | 40.662 | *** | 0.894 | 0.799 | | |
| | OE5 | 0.902 | 0.024 | 37.846 | *** | 0.877 | 0.769 | | |

Note. *** $p$ < 0.001.

**Table 4.** Construct Validity Test Results.

| | AVE | OE | NP | HP | BD | PD |
|---|---|---|---|---|---|---|
| OE | 0.824 | **0.908** | | | | |
| NP | 0.765 | 0.818 | **0.875** | | | |
| HP | 0.745 | 0.796 | 0.762 | **0.863** | | |
| BD | 0.763 | 0.757 | 0.767 | 0.718 | **0.873** | |
| PD | 0.809 | 0.730 | 0.694 | 0.738 | 0.709 | **0.899** |

Note. The boldface in the table is the square root of the AVE of the latent variable and the lower triangle is the Pearson correlation coefficient between the latent variables. AVE = average variance extracted; OE = online brand engagement; NP = novelty perception; HP = hedonic perception; BD = brand image differences; PD = product type differences.

### 4.3. Structural Model Test Results

Model fit tests were performed on the measurement models using Amos 24.0. The model fit indices were within acceptable limits ($\chi^2$ = 272.435; $\chi^2$/df = 1.892; $p$ = 0.000; GFI = 0.950; AGFI = 0.934; CFI = 0.988; TLI = 0.985; RMSEA = 0.041), indicating a good fit of the structural equation model. Furthermore, a method controlling for non-measurable latent method factors was used to test the sample data for common method bias by adding the common method factor as a latent variable to the structural equation model and comparing the change in model fit after adding this latent variable [86]. The results showed that the change in each fit index was less than 0.05 ($\Delta$CFI = 0.005, $\Delta$TLI = 0.004, $\Delta$RMSEA = 0.004, and $\Delta$SRMR = 0.023), and hence, there was no serious problem of common method bias.

The results of the model path test are shown in Figure 2 and Table 5. First, both product type differences ($\beta$ = 0.563; $p$ < 0.001) and brand image differences ($\beta$ = 0.282; $p$ < 0.001) of fashion brands' crossover alliances have a significant positive effect on consumers' novelty perceptions, and the effect of product type differences is greater than that of brand image differences; hypotheses H1 and H2 are supported. Second, both product type differences ($\beta$ = 0.206; $p$ < 0.001) and brand image differences ($\beta$ = 0.342; $p$ < 0.001) of fashion brands' crossover alliances had a significant positive effect on consumers' hedonic perceptions, and the effect of brand image differences was greater than that of product type differences; hypotheses H3 and H4 were supported. Next, consumer novelty perception ($\beta$ = 0.383; $p$ < 0.001) had a significant positive effect on hedonic perception, and hypothesis 5 was supported. Finally, consumer novelty perception ($\beta$ = 0.549; $p$ < 0.001) and hedonic perception ($\beta$ = 0.430; $p$ < 0.001) had a significant positive effect on consumers' online brand engagement. Thus, hypothesis H3 was supported.

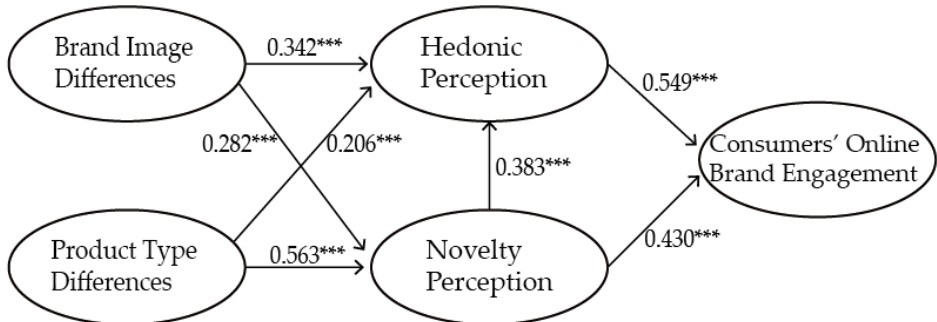

**Figure 2.** Model path analysis results. Note. *** $p$ < 0.001.

**Table 5.** Results of the hypothesis test.

|  | Std. | S.E. | C.R. (t-Value) | $p$ | Hypothesis | Results |
|---|---|---|---|---|---|---|
| PD→NP | 0.563 | 0.051 | 11.040 | *** | H1 | Support |
| BD→NP | 0.282 | 0.043 | 6.510 | *** | H2 | Support |
| PD→HP | 0.206 | 0.060 | 3.403 | *** | H3 | Support |
| BD→HP | 0.342 | 0.046 | 7.422 | *** | H4 | Support |
| NP→HP | 0.383 | 0.059 | 6.471 | *** | H5 | Support |
| NP→OE | 0.549 | 0.048 | 11.374 | *** | H6 | Support |
| HP→OE | 0.430 | 0.045 | 9.527 | *** | H7 | Support |

Note. *** $p$ < 0.001. OE = online brand engagement; NP = novelty perception; HP = hedonic perception; BD = brand image differences; PD = product type differences.

### 4.4. Results of the Mediating Effect Analysis

The Bootstrap method was used for mediating effects analysis, which was tested using the PROCESS macro in SPSS 22.0. The results are shown in Table 6. Firstly, the product type differences in the crossover alliances of fashion brands on social media brand engagement through the mediating effects of novelty perception and hedonic perception

are 0.219 and 0.140, respectively. With a 95% confidence interval not containing 0, this result indicates that the mediating effects are all significant. Secondly, the brand image differences in the crossover alliances of fashion brands on social media brand engagement through the mediating effects of novelty perception and hedonic perception are 0.148 and 0.167, respectively. With a 95% confidence interval not containing 0, this result indicates that the mediating effects are all significant.

**Table 6.** Results of mediating effect test.

| | Total Effect | | Direct Effect | | Indirect Effect Bootstrap 1000 Times 95% CI | | |
|---|---|---|---|---|---|---|---|
| | β | t-Value | β | t-Value | β | BootLLCI | BootULCI |
| PD→NP→OE | 0.455 | 12.277 | 0.236 | 6.395 | 0.219 | 0.144 | 0.313 |
| PD→HP→OE | 0.455 | 12.277 | 0.315 | 8.668 | 0.140 | 0.244 | 0.387 |
| BD→NP→OE | 0.391 | 11.150 | 0.243 | 7.368 | 0.148 | 0.092 | 0.223 |
| BD→HP→OE | 0.391 | 11.150 | 0.223 | 6.243 | 0.167 | 0.109 | 0.244 |

Note. OE = online brand engagement; NP = novelty perception; HP = hedonic perception; BD = brand image differences; PD = product type differences.

## 5. Conclusions and Theoretical Contributions

Based on the S-O-R theoretical model, this paper explores the study of the impact of fashion brands' crossover alliances on consumers' online brand engagement. The results of the analysis based on the sample data of 526 consumers show that the seven hypotheses proposed in this study are all supported. This study has theoretical and managerial implications for the development of crossover alliances of fashion brands with large variations.

### 5.1. Discussion of Conclusions

First, this study reveals that a crossover alliance of fashion brands with distinctiveness can stimulate consumers' perception of novelty to some extent. This finding is consistent with the study of Chun et al. (2015) [87], where brand extensions are likely to increase consumer liking for brand extensions when they have a low fit and provide innovation benefits while having a positive spillover effect on the parent brand. Interestingly, the effect of product type differences on consumers' perceived novelty in this study was twice as high as that of brand image differences (0.563 > 0.282). This result may be because, in general, fashion brands with product type differences have a greater span of crossover alliance, which provokes and attracts consumers to process innovative crossover allied brand messages more deeply, resulting in novelty perception. Meanwhile, fashion brands with brand image differences have a greater span of crossover alliance between fashion brands and novelty perception is smaller.

Second, this study shows that both product type differences and brand image differences have a positive impact on consumers' hedonic perceptions. The path coefficient of the effect of brand image difference on consumers' hedonic perceptions is greater (0.206 < 0.342) compared to product type difference, indicating that the effect of brand image difference is greater. This may be because the brand image difference in the crossover alliance of fashion brands is also mostly a "strong joint" brand cooperation, which tends to make consumers feel a sense of achievement and pride, thus generating hedonic perceptions. In contrast, the difference in product type between fashion brands' crossover alliances is more about creating innovative products that inspire consumers' perception of novelty and thus influence the hedonic perception.

Finally, regarding the antecedents affecting online brand engagement, the findings of this study suggest that both novelty perception and hedonic perception have a significant impact on consumers' online brand engagement. These results confirm the findings of Japutra et al. (2022) [81]. New products that integrate across borders are shared and discussed more, thus making consumers more active in searching for information and increasing

their share of information on social media when novelty perceptions are promoted, and when hedonic perceptions are promoted, consumers are more willing to show and share a satisfying purchase experience on social media.

*5.2. Theoretical Contributions*

First, this study focuses on the emerging industry of crossover alliances of fashion brands. Previous studies related to brand association have focused more on the food, digital electronics, and service industries [88,89], and less research has been conducted on fashion brands' crossover alliances in the related marketing field. Fashion brand companies need to continuously release creative products to attract consumers, and as a result, it is increasingly common for companies to choose crossover alliances, but existing research has not yet explained this phenomenon. Furthermore, studies related to brand extension themes emphasize the critical role of high fit for brand extension success and suggest that low consistency or similarity can lead to negative consumer reactions [90–92]. Although previous studies have also proposed that less consistent new products may elicit curiosity in the context of product rumors [90] and that innovative marketing strategies mitigate the negative effects of crossover alliance evaluations [93], further research on the antecedents of consumer reactions to brand crossover alliances needs to be developed. In contrast, this study, based on the perspective of crossover alliances, divides firms that perform fashion brands crossover alliances into two dimensions: product type differences and brand image differences. It reveals that a brand alliance with low fit also has a positive impact, thus enriching the research related to brand extensions. Moreover, it also provides the means by which fashion brands carry out crossover alliances that can be used to promote consumer brand online engagement in social media.

Second, this study expands the research on consumer responses to brand crossover alliances, especially in terms of social media or the so-called online brand engagement. This result is achieved by introducing novelty perception and hedonic perception into the study, while the study by Giakoumaki and Krepapa (2020) [83] showed that original to interesting content is a driver of online media virality and engagement. However, the impact of fashion brands' extended inconsistency (i.e., crossover alliance) in the social media context has not been previously tested. This research shows that fashion brands' crossover alliances influence consumer online engagement on social media through novelty perception and hedonic perception. The findings also explain, to some extent, the theoretical mechanisms behind the phenomenon of consumers' pursuit of fashion brands' crossover allied products and social media interactions in reality.

## 6. Management Implications

Creating crossover alliances of apparel brands in a manner that stands out from the competition, attracts consumers' attention, and gains widespread consumer engagement in social media is a significant challenge for companies. For that reason, crossover alliances in the fashion sector should be attempted with a limited degree of span, looking for partners with a degree of differentiation from their brand image and main product type. This will stimulate consumers' perception of novelty and hedonism. Fashion products themselves seek to bring different degrees of diversity and novelty to customers. The novelty brought by the crossover marketing of fashion brands can also cater to the new generation of consumers' pursuit of uniqueness and novelty to a certain extent. By joining forces with brands that do not compete with each other in their own business racing tracks, they can also expand their fan base and inject new connotations into the brand itself.

In addition, the novelty and a heightened sense of hedonism of crossover allied fashion brands can capture the new generation of consumers who are eager to express themselves on social media. This will not only share traffic between the co-brandings in social networks but also stimulate other consumers to become participants and co-creators of brand traffic, helping brands to build awareness online and attract potential customers through existing ones. For illustration, when LV and BMW joined forces to co-launch the i8 sports car

four-piece travel luggage collection for luxury goods, both brands published an alliance message on their respective social media pages. Their actions generated a huge response on social media, where consumers created their personas and were inspiring new customers for the brand. Brands with multiple followers based on social media can build brand communities, maintain closer relationships between consumers and brands, meaningfully communicate brand values, and achieve more accurate marketing with fewer costs.

### 7. Limitation and Future Expectations

First, this study was conducted by means of a web-based questionnaire. The study selected six cases of crossover alliances of fashion brands that have occurred in recent years and are more typical and well-known, in order for consumers to understand the concept of crossover alliances of fashion brands. In the research process, however, some emotional factors such as consumers' familiarity with the brands may interfere. In future studies, longitudinal data and experimental methods can therefore be used to verify the conclusions of this paper.

Second, this study is conducted for fashion brand crossover alliances in the apparel industry, which is a low-involvement product. However, whether crossover alliances based on high-involvement products can still cause consumers to engage with brands online needs further tests in future research.

Third, as there is a large cross-border contrast in fashion brand crossover alliances, which may also have negative spillover effects, this issue can also be analyzed in depth in further studies.

**Author Contributions:** Conceptualization, J.C.; methodology, J.C.; software, J.W.; validation, J.W.; formal analysis, J.C.; investigation, J.C. and Y.C.; data curation, J.C. and J.W.; writing—original draft preparation, J.C.; writing—review and editing, J.C. and H.Z.; funding acquisition, H.Z. All authors have read and agreed to the published version of the manuscript.

**Funding:** This research was funded by Fujian Province higher education for technology innovation team (2018); Fujian Provincial Department of Education (No. JAT190528); Fujian Provincial Department of Education Project (No. FBJG20220212).

**Institutional Review Board Statement:** Not Applicable.

**Informed Consent Statement:** Not Applicable.

**Data Availability Statement:** Not Applicable.

**Acknowledgments:** We would like to thank the anonymous referees for commenting on this paper.

**Conflicts of Interest:** The authors declare no conflict of interest.

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
