# Peer review of "Research on the Influence Mechanism of Fashion Brands’ Crossover Alliance on Consumers’ Online Brand Engagement: The Mediating Effect of Hedonic Perception and Novelty Perception"

_sustainability, doi:10.3390/su15053953_

Round 1

Reviewer 1 Report

I would first like to congratulate the authors for their quality effort in planning, reviewing and formulating the empirical exercise. The topic of online brand engagement in perceptual contexts is extremely interesting, current, and of enormous importance for actors in e-marketplaces. At the theoretical level, the research work touches on the main contemporary authors, which allowed the formulation of a series of hypotheses around a theoretical model. It abundantly describes the importance of each hypothesis, where cumulatively it correctly describes the methodology of the empirical study. Both the discussion and the results presented have scientific relevance, which pleases any reviewer of such a paper.

Author Response

Thank you for your careful reading of our manuscript. 

Thank you for your recognition and we will continue to work on it.

Best wishes,

Jin jiang Cai

Reviewer 2 Report

The paper pretends to be based on a study that introduce two mediating variables, novelty perception, and hedonic perception, to explore the potential mechanismof brandimage21 differences and product type differences on consumers' online brand engagement underthe22 background of crossover alliance of fashion brands.

The Introduction makes general claims that are not supported neither by theoretical marketing literature, nor by recent studies in the domain. The choice of some specific luxury brands to clarify the claims is not justified. The literature review is sporious and poorly written. Most of the theoretical ideas pretended to be marketing knowledge  are not in line with accepted body of knowledge neither with recent studies. The authors do not properley know the concepts from marketing literature, therefore their use is improper. The Introduction does not specify or explain in a convincing way why the study is necessary (what is the problem adressed), nor what is the knowledge gap that it tries to bridge.

Theoretical framework seems made-up, as the chosen concepts do not allingn with the marketing literature; moreover it is badly written (it needs to be synthetised). The claims proposed as theoretical knowledge are exagerated and not credible.

The entire paper is written in the same way which makes it questionable (the authors do not seem to have scientific knowledge in the field, not to base their paper on an actually conducted study).

Author Response

RESPONSES TO REVIEWER # 2

Dear Reviewer,

    Thank you very much for the reviewer’ comments concerning our manuscript entitled “ Research on the Influence Mechanism of Fashion Brands’ Crossover Alliance on Consumers’ Online Brand Engagement: The Mediating Effect of Hedonic Perception and Novelty Perception ” (ID: 2093483). These comments are very valuable and helpful for revising and improving our paper.  

    We have studied the valuable comments from you carefully, and tried our best to revise the manuscript, which we hope meet the approval. Revised portion has been marked in red through the paper.

    The point to point responses to the reviewer’s comments are listed as following:

Comment 1
The Introduction makes general claims that are not supported neither by theoretical marketing literature, nor by recent studies in the domain. The choice of some specific luxury brands to clarify the claims is not justified.  

Authors’ response

Thank you for your careful reading of our manuscript. In the revised manuscript, we have rewritten the Introduction. For your convenience, we have included the section below:

1 Introduction

    The rapid development of digital technology has made social media popular these days. Various social media sites have penetrated and become an integral part of consumers' daily lives, while companies are increasingly focusing on and valuing the ability of social media to contribute to their businesses [1]. More and more enterprises are using social media to post content related to their products in order to generate consumer excitement and increase brand engagement (likes, shares, comments, etc.) in the business competition [2]. Especially for products with high attention and low engagement such as apparel, when many fashion brands try to take a crossover alliance approach to launch new products, it is necessary to attract consumers' attention and motivate them to participate in brand-related activities online to have more awareness and influence. For example, Dior and Air Jordan 1 High OG collaborated on a co-branded sneaker that attracted great attention in social media and fetched a high price of tens of thousands of dollars on second-hand platforms; Gucci and The North Face crossover alliance, once the preview has caused a buzz in social networks; Balenciaga's crossover collaboration with Fortnite the online game, which simultaneously launched online game virtual sets and offline physical clothing, has achieved excellent business performance. The above cases show that the crossover alliances of fashion brands have gained good market feedback, both in the offline consumer market and online stream. However, it is worth noting that there are also some cases of crossover alliances that have had the opposite effect. For example, the joint H&M and Kenzo model was hailed by netizens as "the most difficult to wear collection ever". The ZX 7000 sneakers jointly launched by HEYTEA and Adidas Originals were also criticized on social media. Consequently, how enterprises can develop effective crossover alliance strategies to help fashion brands gain market share and enhance consumer brand engagement on social media is a critical issue that needs to be addressed.

    In the apparel and fashion industry, brand association strategy has been one of the preferred ways for companies to explore new market values and gather a customer base. Brand association is a marketing strategy in which two or more similar brands cooperate to create new products [3]. The associations of fashion brands combine the respective characteristics of the constituent brands and transfer the relevant values into the co-branded products [4]. The aim is to drive discovery and familiarity with branded products among consumers who are not aware of the brand. Previous studies on the topic of brand alliances have given extensive attention, mostly focusing on the fast moving consumer goods and electronics markets[5], while only some studies have focused on fashion brand companies, analyzing fashion brand business model innovation [6], consumer brand loyalty[7], and the impact of fashion brand alliances on brand equity [4,8]. Scholars have emphasized the key factors of successful brand association based on the concept of "brand association similarity" [9], and have shown that similarity in brand image, product category, product attributes, and product quality among the collaborating brands can trigger consumer perception of fit [10-15], and higher perception of fit has a positive impact on brand evaluation [3]. However, mechanisms based on co-branding similarity and perceived fit do not explain well the existing successful cases of brand crossover cooperation in fashion marketing.

    It is well known that the fashion industry has always been a competitive and fast-changing market, and fashion brands need to continuously push the boundaries to ensure market growth, enhance consumer recognition and ultimately maintain a competitive advantage [16], but which model of fashion brands’ crossover alliance influences consumer perception and engagement remains unclear and requires further research. In addition, research has shown that consumer online brand engagement is an important strategic factor [17]. This is because companies channel offline activities and online streams into cyber environments, where they can track consumers' digital footprints more accurately and further delve into customer preferences [18]. However, existing research on fashion brand alliance only focuses on the effects on consumers' behavioral decisions such as purchase intention [19,20], payment premium [21], and willingness to recommend to others [22]. In the meantime, the understanding of consumers' online brand engagement remains limited, and little exploration has been conducted on the relationship between fashion brand crossover alliance and consumers' online brand engagement.  

Comment 2

The literature review is sporious and poorly written. Most of the theoretical ideas pretended to be marketing knowledge  are not in line with accepted body of knowledge neither with recent studies. The authors do not properley know the concepts from marketing literature, therefore their use is improper.  

Authors’ response

We have rewritten this part carefully, reorganized the relevant literature, and added more relevant and recent literature. For your convenience, we have included the section below:

2. Theoretical backgrounds

2.1. Crossover Alliance of Fashion Brands

    Brand association originates from the concept of symbiotic marketing [9], which refers to an effective way for two or more brands to form a short-term or long-term collaboration with the aim of creating synergies and complementing each other's resources to broaden the market and increase the brand value [3,25-27]. The number of business practices and academic research literature on brand alliances has increased significantly over the past few years, as companies have become increasingly aware of the growth opportunities offered by brand alliance strategies [28]. Previous studies have focused on exploring the impact of factors such as consumers' brand loyalty, perceived brand quality, and brand fit on brand alliance success [5,29]. It is emphasized in the study that among brand association partners, either brand image fit, product category fit, or consumer perception fit affects consumer perception fluency and further influences brand association attitudes [8,30]. However, with the evolution of the business market, high fit brand alliances in the field of fashion marketing are no longer sufficient to meet market and consumer needs. For instance, Moon and Sprott (2016) [31] showed that luxury brands may find it almost impossible to collaborate with other luxury brands and therefore need to consider non-luxury brand partners. Co-partners with too similar brand images or product types may struggle to inspire freshness in consumers and thus reduce purchase intentions [32]. Further, it has been emphasized that moderate dissonance has a positive and interesting value. Studies Srivastava and Sharma (2012) [33] and Walchli (2007) [34] pointed out that in the brand extension process, the degree of brand extension consistency shows an inverted U-shaped relationship with brand extension evaluation, i.e., moderate incongruence leads to better brand evaluation than homogeneity or extreme incongruence. Based on product innovation domain theory, Zhu and Wang (2016) [35] emphasizes the importance of consumer novelty perceptions in the brand innovation process. Consumers may have stronger novelty perceptions and hedonic perceptions for innovative products that deviate from the norms. These phenomena may be due to the possibility that consumers find the brand or product more unique, iconic, or stylish [36,37].

    As a result, from luxury brands to fast fashion brands, all companies try to break the original brand image, seek differentiation, and create a "brand crossover alliance" [38]. Brand crossover alliance is a form of brand alliance, which is defined by scholars as the joint development of new products by two or more brands in non-competing industries that highly integrate the characteristics or attributes of the joint brand [39]. Further, some scholars argue that "crossover" cannot be simply summarized as the difference between industries or sectors in an objective sense. The so-called crossover should be a brand association partner that consumers perceive to be less related, similar, compatible, or complementary in terms of their salient attributes [40]. In this way, a crossover alliance between fashion brands can be a collaboration between two brands with widely different images in the fashion industry. For example, Gucci and The North Face are both fashion brands, but in terms of brand image, one is a luxury brand and the other is an outdoor sports brand. The perceived similarity of their brand images is rather small, and the product positioning or product values are very different. Oeppen and Jamal (2014) [41] and Mrad et al. (2019) [42] explored through qualitative research the good results achieved by fast fashion brands and luxury brands through low-scale collaborations. This is because it seems that through short-term collaborations luxury brands can be protected from being viewed negatively by their existing consumer base, while also attracting potential consumers to new markets through the mass market of fast fashion brands. On the other hand, the crossover alliance of fashion brands can also be the collaboration of the fashion industry with other industries. For example, Balenciaga and Moncler have successively designed virtual clothing with the online game Fortnite. In terms of product type, one is an Internet game company and the other is a luxury fashion brand, and both product types have strong differences in terms of product attributes and product usage scenarios. The case study by Moon and Sprott (2016) [31] shows that when fashion brands and brands from other sectors unite, consumers' perceived brand image fit positively influences purchase intentions for co-branded new products. The study by Paydas (2021) [21] shows that brand alliances of different product types positively influence consumers' perceptions of product quality and enhance willingness to purchase the product.

    Reviewing the above literature, we find that scholars analyze the effectiveness of fashion brand crossover alliances from the perspective of brand image difference or product type difference, respectively, but fewer scholars incorporate the two dimensions into one model for exploration and comparison. Thus, this study explores fashion brand crossover alliances from two dimensions: brand image difference and product type difference, and defines fashion brand crossover alliances as the joint development of new products by a fashion brand with single or multiple brands that have outstanding differences at the brand image level or product type level. Differences at the brand image level refer to consumers' perceptions that the brand image, brand positioning and brand perception of the partners are not similar; differences at the product type level refer to consumers' perceptions that the product attributes and product usage contexts of the partners are not compatible and complementary.

2.2. Online Brand Engagement of Consumers

    The concept of consumer online brand engagement is derived from the service-dominant logic of Relationship Marketing Theory [43], but there is still no consensus among academicians on the definition of the concept. Engagement is a multifaceted concept that encompasses a range of cognitive, affective, and behavioral dimensions [17]. Brand-related online behaviors such as browsing, consuming, commenting, and sharing can be conceptualized as online brand engagement [44]. Muntinga et al. (2021) [45] dissected three structures of consumers’ online brand engagement in conjunction with the COBRAs framework. One is "consumption", where consumers may only engage but not post content, such as watching videos or reading reviews. This category is also referred to as "passive users" [46]. The second is "contribution", where consumers may join brand communities to communicate and discuss, or comment on relevant content posted by brands. The third is "creation," in which consumers create brand-related content that is viewed and discussed by other participants of the brand. For example, consumers are transformed into active brand participants [17] and creators of brand stories [47] through social networks. Similarly, Khan (2017) [46] divided consumer online brand engagement behavior into two structures, namely consumption and engagement. Consumption includes consumers watching and reading branded content, while engagement includes consumer likes, feedback, recommendations, and conversations. Since this study aims to understand consumers' reactions to new products combined by brands across borders, it focuses on consumer engagement, defining consumer online brand engagement as consumers expressing likes, sharing comments, and recommending purchases of apparel (or products) combined by fashion brands across borders on social media.

    The large user base of social media (e.g., Facebook, Twitter, Instagram, and LinkedIn) has brought its importance to the attention of more brands [48]. The interactive, participatory and open nature of social media provides an easier and more effective channel for brands to discover consumer needs and build connections [49]. As the market becomes more competitive, brands want to build close connections with consumers through various online marketing promotions, rather than simply a single purchase relationship [50,51]. Therefore, seeking consumer online brand engagement has become of growing importance to companies. The existing literature suggests that either intrinsic consumer-based perceptions or external cue-based factors are key antecedents influencing consumer online brand engagement. Based on intrinsic consumer perceptions, factors such as consumer satisfaction, trust, brand commitment, brand attachment, and perceived brand performance are important factors influencing consumer engagement [52]. But most of them are explored based on traditional brand marketing scenarios, and there may be different key drivers such as novelty perception and hedonic perception in brand crossover alliance scenarios. Studies by Zhu et al. (2016) and Dun et al. (2020) [35,53] both emphasize the importance of novelty perception in the brand product innovation process, which has a significant impact on consumers' purchase intentions. In addition, Kim et al. (2023) [24] showed that hedonic perception has a significant role between brand advertising interactivity and consumers' purchase intention. Despite the fact that scholars have explored the influence of novelty perception and hedonic perception on consumers' purchase decisions, there is a need to further understand the impact of novelty perception and hedonic perception on consumers' online brand engagement in the context of crossover alliance from the perspective of fashion companies. Based on the external cues aspect, consumers are more likely to engage with social media content that introduces new products with low consistency [54]. In line with this, Borah et al. (2020) [55] suggests that topical, timely or unexpected improvised content items can facilitate consumer conversations with brands on social media. Lee et al. (2018) [56] states that content related to brand personality (e.g., humor and emotion) inspires higher levels of consumer engagement. Thus, product content that differs from traditional brand associations, such as new product content from fashion brands associated across borders, may be appropriate for the social media environment, but the impact of brand crossover alliances on consumer online brand engagement and its underlying mechanisms need to be further investigated.

Comment 3

The Introduction does not specify or explain in a convincing way why the study is necessary (what is the problem adressed), nor what is the knowledge gap that it tries to bridge.

Authors’ response

According to reviewer’s comment, we have elaborated the theoretical gaps and possible contributions more clearly from paragraph 4 in the Introduction section. The changes have also been marked in red in the paper. For your convenience, we have included the section below:

    To settle the above issues, this study has two objectives: first, this study divides fashion brand crossover alliances into brand level and product level--in other words, brand image difference and product type difference--and argues that brand alliances with low fit can also have positive impacts, thus enriching the research related to brand extensions. The second objective is to clarify how fashion brand crossover alliances affect consumers' online brand engagement. Specifically, using the Stimulus-Organism-Response (S-O-R) theory, two mediating variables, consumer novelty perception and hedonic perception, are introduced to explore the potential mechanisms by which brand image differences and product type differences in fashion brand crossover alliances affect the formation of consumer online brand engagement. In terms of industry characteristics, consumer brand engagement on social media is particularly important for fashion brands [23,24]. Although consumer online brand engagement may contribute to the formation of relationships such as familiarity, loyalty, and trust [18], there is less research in the existing literature on the relationship between crossover alliances of fashion brands and consumer online brand engagement. This study enriches the antecedent theory of consumer online brand engagement and also provides a reference for companies to carry out fashion brand crossover alliances to enhance social media engagement.

Comment 4
Theoretical framework seems made-up, as the chosen concepts do not allingn with the marketing literature; moreover it is badly written (it needs to be synthetised). The claims proposed as theoretical knowledge are exagerated and not credible.

Authors’ response

Following this suggestion, we have reorganized the relevant literature and theories, added relevant research and analysis, and ensured that our theoretical framework is more rational. For your convenience, we have included the section below: 

2.3. Stimulus—Organism—Response (S-O-R) Theory

    Mehrabian & Russell (1974) introduced the "organism" factor from the perspective of environmental psychology. They studied the internal response of individuals to environmental stimuli and proposed the "Stimulus-Organism- Response" theory. The S-O-R model states that individuals' behavioral decisions are not always rational and that they may be influenced by external stimuli (e.g., products or situations) (S) that affect their internal perceptions, cognition, or subconscious (O) and drive behavioral responses (R) [57]. The use of the S-O-R model allows a better connection between external stimuli and consumers' internal perceptions and reactions [36] and has been widely used in online social media scenarios to analyze consumer responses and behavioral preferences [24,58]. For example, Casaló and Ibáñez-Sánchez (2021) [58] found that the content (S) posted by fashion brands in social media influences consumers' perceived creativity and positive emotions (O) toward the brand, which increases emotional commitment and willingness to interact (R) with the brand. Kim et al. (2023) [24] found that the interactivity (S) of fashion brand advertising triggers consumers' hedonic perceptions (O) and increases consumers' purchase (R).

    While previous studies have used branded online advertising as a stimulus, they have not examined how specific features of brand advertising (e.g., the particular product message of fashion brand crossover alliances) impact consumer perceptions and behaviors. Therefore, this study extends the S-O-R framework to the domain of fashion brand crossover alliances research. For a peripheral stimulus cue (i.e., fashion brand crossover association), the S-O-R framework is used to explore the impact of fashion brand crossover alliance on consumers' online brand engagement. The brand image difference and product type difference of fashion brand crossover alliance are used as external environment stimulus variables (S), hedonic perception and novelty perception as individual cognitive psychological variables (O), and consumer online brand engagement as the driving behavioral response (R).

Once again, thank you very much for your valuable comments and detailed suggestions. And we hope our correction will meet the approval.  

References

  1. Guercini, S.; Bernal, P.M.; Prentice, C. New marketing in fashion e-commerce. J. glob. fash. mark. 2018, 9, 1-8, doi:10.1080/20932685.2018.1407018.
  2. Liu, Y.; Liu, X.; Wang, M.; Wen, D. How to catch customers’ attention? A study on the effectiveness of brand social media strategies in digital customer engagement. Front. psychol. 2021, 12, doi:10.3389/fpsyg.2021.800766.
  3. Ahn, J.; Kim, A.; Sung, Y. The effects of sensory fit on consumer evaluations of co-branding. Int. j. advert. 2019, 39, 486-503, doi:10.1080/02650487.2019.1652518.
  4. Yu, Y.; Rothenberg, L.; Moore, M. Exploring young consumer's decision‐making for luxury co-branding combinations. Int. j. retail. distrib. 2021, 49, 341-358, doi:10.1108/IJRDM-12-2019-0399.
  5. Paydas Turan, C. Success drivers of co-branding: A meta-analysis. Int. j. consum. stud. 2021, 45, 911-936, doi:10.1111/ijcs.12682.
  6. Bai, H.; McColl, J.; Moore, C. Luxury fashion retailers' localised marketing strategies in practice – evidence from China. Int. market. rev. 2022, 39, 352-370, doi:10.1108/IMR-02-2021-0079.
  7. Shen, B.; Choi, T.-M.; Chow, P.-S. Brand loyalties in designer luxury and fast fashion co-branding alliances. J. bus. res. 2017, 81, 173-180, doi:10.1016/j.jbusres.2017.06.017.
  8. Ma, B.; Cheng, F.; Bu, J.; Jiang, J. Effects of brand alliance on brand equity. Journal of Contemporary Marketing Science 2018, 1, 22-33, doi:10.1108/JCMARS-08-2018-0007.
  9. Adler, L. Symbiotic marketing. Harvard. bus. rev. 1966, 44, 59-71.
  10. Aaker, D.A.; Keller, K.L. Consumer Evaluations of Brand Extensions. J. marketing. 1990, 54, 27-41, doi:10.1177/002224299005400102.
  11. Xiao, N.; Hwan Lee, S. Brand identity fit in co-branding: The moderating role of CB identification and consumer coping. Eur. j. marketing. 2014, 48, 1239-1254, doi:10.1108/EJM-02-2012-0075.
  12. Lee, J.K.; Lee, B.-K.; Lee, W.-N. Country-of-origin fit's effect on consumer product evaluation in cross-border strategic brand alliance. J. bus. res. 2013, 66, 354-363, doi:10.1016/j.jbusres.2011.08.016.
  13. Bhat, S.; Reddy, S.K. The impact of parent brand attribute associations and affect on brand extension evaluation. J. bus. res. 2001, 53, 111-122, doi:10.1016/S0148-2963(99)00115-0.
  14. Keller, K.L. Conceptualizing, Measuring, and Managing Customer-Based Brand Equity. J. marketing. 1993, 57, 1-22, doi:10.1177/002224299305700101.
  15. Cornelis, P.C.M. Effects of co‐branding in the theme park industry: a preliminary study. Int. j. contemp. hosp. m. 2010, 22, 775-796, doi:10.1108/09596111011063089.
  16. Gao, L.; Norton, M.J.T.; Zhang, Z.m.; Kin‐man To, C. Potential niche markets for luxury fashion goods in China. J. fash. mark. manag. 2009, 13, 514-526, doi:10.1108/13612020910991376.
  17. Hollebeek, L.D.; Glynn, M.S.; Brodie, R.J. Consumer Brand Engagement in Social Media: Conceptualization, Scale Development and Validation. J. interact. mark. 2014, 28, 149-165, doi:10.1016/j.intmar.2013.12.002.
  18. Wang, T.; Lee, F.-Y. Examining customer engagement and brand intimacy in social media context. J. retail. consum. serv. 2020, 54, 102035, doi:10.1016/j.jretconser.2020.102035.
  19. Wu, D.G.; Chalip, L. Effects of co-branding on consumers' purchase intention and evaluation of apparel attributes. J. glob. scholars. mark. 2014, 24, 1-20, doi:10.1080/21639159.2013.852910.
  20. Mazodier, M.; Merunka, D. Beyond brand attitude: Individual drivers of purchase for symbolic cobranded products. J. bus. res. 2014, 67, 1552-1558, doi:10.1016/j.jbusres.2014.01.015.
  21. Paydas Turan, C. What's inside matters: The impact of ingredient branding on consumers' purchasing behaviours in services. J. retail. consum. serv. 2021, 63, 102690, doi:10.1016/j.jretconser.2021.102690.
  22. Ho, H.-C.; Lado, N.; Rivera-Torres, P. Detangling consumer attitudes to better explain co-branding success. J. Prod. Brand. Manag. 2017, 26, 704-721, doi:10.1108/JPBM-11-2015-1039.
  23. Molina-Prados, A.; Muñoz-Leiva, F.; Prados-Peña, M.B. The role of customer brand engagement in the use of Instagram as a “shop window” for fashion-industry social commerce. J. fash. mark. manag. 2022, 26, 495-515, doi:10.1108/JFMM-12-2020-0275.
  24. Kim, K.; Chung, T.-L.; Fiore, A.M. The role of interactivity from Instagram advertisements in shaping young female fashion consumers’ perceived value and behavioral intentions. J. retail. consum. serv. 2023, 70, 103159, doi:10.1016/j.jretconser.2022.103159.
  25. Rao, A.R.; Ruekert, R.W. Brand alliances as signals of product quality. Mit. sloan. manage. rev. 1994, 36, 87-97.
  26. Rao, A.R.; Qu, L.; Ruekert, R.W. Signaling Unobservable Product Quality through a Brand Ally. J. marketing. res. 1999, 36, 258-268, doi:10.1177/002224379903600209.
  27. Newmeyer, C.E.; Venkatesh, R.; Ruth, J.A.; Chatterjee, R. A typology of brand alliances and consumer awareness of brand alliance integration. Market. lett. 2018, 29, 275-289, doi:10.1007/s11002-018-9467-4.
  28. Newmeyer, C.E.; Venkatesh, R.; Chatterjee, R. Cobranding arrangements and partner selection: a conceptual framework and managerial guidelines. J. acad. market. sci. 2014, 42, 103-118, doi:10.1007/s11747-013-0343-8.
  29. Eren-Erdogmus, I.; Akgun, I.; Arda, E. Drivers of successful luxury fashion brand extensions: cases of complement and transfer extensions. J. fash. mark. manag. 2018, 22, 476-493, doi:10.1108/JFMM-02-2018-0020.
  30. Ahn, J.; Kim, A.; Sung, Y. The effects of sensory fit on consumer evaluations of co-branding. Int. j. advert. 2020, 39, 486-503, doi:10.1080/02650487.2019.1652518.
  31. Moon, H.; Sprott, D.E. Ingredient branding for a luxury brand: The role of brand and product fit. J. bus. res. 2016, 69, 5768-5774, doi:10.1016/j.jbusres.2016.04.173.
  32. Sénéchal, S.; Georges, L.; Pernin, J.L. Alliances Between Corporate and Fair Trade Brands: Examining the Antecedents of Overall Evaluation of the Co-branded Product. J. bus. ethics. 2014, 124, 365-381, doi:10.1007/s10551-013-1875-z.
  33. Srivastava, K.; Sharma, N.K. Consumer attitude towards brand-extension incongruity: The moderating role of need for cognition and need for change. J. market. manag-uk. 2012, 28, 652-675, doi:10.1080/0267257X.2011.558383.
  34. Walchli, S.B. The effects of between-partner congruity on consumer evaluation of co-branded products. P&M. 2007, 24, 947-973, doi:10.1002/mar.20191.
  35. Zhu, Q.; Wang, X.-y. The Influential mechanism of perceived product Innovativeness on consumers’purchase intention: The moderating role of country-of-origin image and price sensitivity. Business and Management Journal 2016, 38, 107-108, doi:10.19616/j.cnki.bmj.2016.07.010.
  36. Warren, C.; Batra, R.; Loureiro, S.M.C.; Bagozzi, R.P. Brand Coolness. J. marketing. 2019, 83, 36-56, doi:10.1177/0022242919857698.
  37. De Veirman, M.; Cauberghe, V.; Hudders, L. Marketing through Instagram influencers: the impact of number of followers and product divergence on brand attitude. Int. j. advert. 2017, 36, 798-828, doi:10.1080/02650487.2017.1348035.
  38. Sharma, A.; Soni, M.; Borah, S.B.; Saboo, A.R. Identifying the drivers of luxury brand sales in emerging markets: An exploratory study. J. bus. res. 2020, 111, 25-40, doi:10.1016/j.jbusres.2020.02.009.
  39. Zhou, Y.; Zhang, Y.; Chen, S. Influence of product association distance on the evaluation of joint product and consumers' purchase intention in the context of brand crossover alliance. Journal of Business Economics 2021, 37-50, doi:10. 14134 /j. cnki. cn33-1336 /f. 2021. 12. 003.
  40. Jian, Y.; Zhu, L.; Zhou, Z. The mechanism of brand alliance attitude towards Cross-boundary Co-development: Based on the Perspectiveof Consumer Inspiration Theory. Nankai. bus. rev. nt. 2021, 24, 25-38, doi:10.3969/j.issn.1008-3448.2021.02.004.
  41. Oeppen, J.; Jamal, A. Collaborating for success: managerial perspectives on co-branding strategies in the fashion industry. J. market. manag-uk. 2014, 30, 925-948, doi:10.1080/0267257X.2014.934905.
  42. Mrad, M.; Farah, M.F.; Haddad, S. From Karl Lagerfeld to Erdem: a series of collaborations between designer luxury brands and fast-fashion brands. J. brand. manag. 2019, 26, 567-582, doi:10.1057/s41262-018-00146-2.
  43. Brodie, R.J.; Hollebeek, L.D.; Jurić, B.; Ilić, A. Customer Engagement: Conceptual Domain, Fundamental Propositions, and Implications for Research. J. serv. res-us. 2011, 14, 252-271, doi:10.1177/1094670511411703.
  44. Eigenraam, A.W.; Eelen, J.; van Lin, A.; Verlegh, P.W.J. A Consumer-based Taxonomy of Digital Customer Engagement Practices. J. interact. mark. 2018, 44, 102-121, doi:10.1016/j.intmar.2018.07.002.
  45. Muntinga, D.G.; Moorman, M.; Smit, E.G. Introducing COBRAs. Int. j. advert. 2011, 30, 13-46, doi:10.2501/IJA-30-1-013-046.
  46. Khan, M.L. Social media engagement: What motivates user participation and consumption on YouTube? Comput. hum. behav. 2017, 66, 236-247, doi:10.1016/j.chb.2016.09.024.
  47. De Vries, N.J.; Carlson, J. Examining the drivers and brand performance implications of customer engagement with brands in the social media environment. J. brand. manag. 2014, 21, 495-515, doi:10.1057/bm.2014.18.
  48. Thaichon, P.; Brown, J.R.; Weaven, S. Special issue introduction: online relationship marketing. Mark. intell. plan. 2020, 38, 673-675, doi:10.1108/MIP-09-2020-623.
  49. Robson, S.; Banerjee, S. Brand post popularity on Facebook, Twitter, Instagram and LinkedIn: the case of start-ups. Online. inform. rev. 2022, ahead-of-print, doi:10.1108/OIR-06-2021-0295.
  50. Kumar, V. Building Customer-Brand Relationships through Customer Brand Engagement. J. Promot. Manag. 2020, 26, 986-1012, doi:10.1080/10496491.2020.1746466.
  51. Lim, W.M.; Kumar, S.; Pandey, N.; Rasul, T.; Gaur, V. From direct marketing to interactive marketing: a retrospective review of the. J. res. interact. mark. 2022, ahead-of-print, doi:10.1108/JRIM-11-2021-0276.
  52. de Oliveira Santini, F.; Ladeira, W.J.; Pinto, D.C.; Herter, M.M.; Sampaio, C.H.; Babin, B.J. Customer engagement in social media: a framework and meta-analysis. J. acad. market. sci. 2020, 48, 1211-1228, doi:10.1007/s11747-020-00731-5.
  53. Dun, S.; Chen, Q.; Xie, Z.-m.; Hu, X.-p. The mechanism of perceived product innocationn on purchase intention-Evidence from the smartphone industry. Systems Engineering 2020, 38, 43-51.
  54. Gerrath, M.H.E.E.; Biraglia, A. How less congruent new products drive brand engagement: The role of curiosity. J. bus. res. 2021, 127, 13-24, doi:10.1016/j.jbusres.2021.01.014.
  55. Borah, A.; Banerjee, S.; Lin, Y.-T.; Jain, A.; Eisingerich, A.B. Improvised Marketing Interventions in Social Media. J. marketing. 2020, 84, 69-91, doi:10.1177/0022242919899383.
  56. Lee, D.; Hosanagar, K.; Nair, H.S. Advertising Content and Consumer Engagement on Social Media: Evidence from Facebook. Manage. sci. 2018, 64, 5105-5131, doi:10.1287/mnsc.2017.2902.
  57. Belk, R.W. Situational variables and consumer behavior. J. consum. res. 1975, 2, 157-164, doi:doi.org/10.1086/208627.
  58. Casaló, L.V.; Flavián, C.; Ibáñez-Sánchez, S. Be creative, my friend! Engaging users on Instagram by promoting positive emotions. J. bus. res. 2021, 130, 416-425, doi:10.1016/j.jbusres.2020.02.014.

Reviewer 3 Report

This study is based on the question of whether the Fashion Brands' Crossover Alliance can effectively increase consumers' online brand engagement. Using the S-O-R model, the article introduces two mediating variables, novelty perception and hedonic perception, to explore the potential mechanism of brand image differences and product type differences on consumers' online brand loyalty under the background of cross-alliance of fashion brands.

There are seven main hypotheses in the study. These hypotheses are as follows:

H1: Brand image differences have a positive impact on consumers' hedonic perceptions in the crossover alliance of fashion brands.

H2: Product type differences have a positive impact on consumers' hedonic perceptions in the crossover alliance of fashion brands.

H3: Brand image differences have a positive impact on consumers' novelty perception in the crossover alliance of fashion brands.

H4: Product type differences have a positive impact on consumers' novelty perception in the crossover alliance of fashion brands.

H5: Novelty perception has a positive impact on consumers' hedonic perception in the crossover alliance of fashion brands.

H6: Novelty perception has a positive impact on consumers’ online brand engagement in the crossover alliance of fashion brands.

H7: Hedonic perception has a positive impact on consumers’ online brand engagement in the crossover alliance of fashion brands.

The contributions of the study to the literature are as follows:

First, unlike most other studies, this study focuses on the emerging cross-alliance industry of fashion brands. Second, this study expands research on consumer responses to cross-brand alliances, particularly in terms of social media or so-called online brand engagement.

The findings of the study are summarized as follows:

First, the study reveals that cross-alliance of fashion brands with distinctive features can stimulate consumers' perception of innovation to some extent. Second, the study shows that both product type differences and brand image differences have a positive effect on consumers' hedonic perceptions. Finally, regarding the antecedents affecting online brand engagement, the findings of this study suggest that both novelty perception and hedonic perception have a significant impact on consumers' online brand engagement.

First of all, I think the subject of the study is interesting and original. I found the work generally successful. The theoretical background of the study is adequately explained. The literature is current and sufficient. The contribution of the study to the literature is clearly written. The findings of the study have been sufficiently discussed. Detailed discussions were made. The results are presented in comparison with the previous literature. The limitations of the study are adequately explained. The necessary statistics were given by applying the analyzes correctly. I did not see any problem in the analysis part. I do not have any correction requests regarding the article.

Author Response

(The authors gave the same response as above.)

Reviewer 4 Report

I’m very grateful for the opportunity to review this manuscript, and I hope that my comments will help the authors to improve the current state of the document.

The introductory section seems completely adequate to me and I don’t need to comment. However, in section 2 (theoretical background), section 2.1 I think can be improved considerably. On the one hand, the authors speak in several paragraphs of the crossed alliance in terms of definition (lines 107-109 and lines 127-129). I believe that this information should be better reorganized. On the other hand, they end section 2.1 by talking about the differences in brand image and type of product, which I consider to be a very important part and therefore could be explored further. According to the authors, there are two dimensions (line 126). What authors or previous studies support these dimensions? Is there specific literature on these dimensions? I have no comment to make on the rest of the sections.

In section 4 (methodology), the authors talk about “questions for each variable adapted from the literature”, what adaptations were necessary? What process was used for adaptation? This tool or any of its latent variables, in what previous studies has it been used? With what psychometric results? It would be interesting to include information on this. Regarding the most well-known alliance cases, is there any specific criteria for using those cases and not others? Regarding the screening question, how many people indicated “no”? (identify how many people indicated “no” and how many people gave incomplete answers). In section 4.2, is it possible to include the CFA adjustment information as well? Use the same indices as for the SEM analysis in section 4.3.

The limitations do not indicate anything about the sample and its composition. Is there any limitation that 75% of the sample is between the ages of 18 and 39?

The references do not conform to the journal's regulations. Please review this section completely.

Author Response

RESPONSES TO REVIEWER # 4

Dear Reviewer,

Thank you very much for the reviewer’ comments concerning our manuscript entitled “ Research on the Influence Mechanism of Fashion Brands’ Crossover Alliance on Consumers’ Online Brand Engagement: The Mediating Effect of Hedonic Perception and Novelty Perception ” (ID: 2093483). These comments are very valuable and helpful for revising and improving our paper.  

We have studied the valuable comments from you carefully, and tried our best to revise the manuscript, which we hope meet the approval. Revised portion has been marked in red through the paper.

The point to point responses to the reviewer’s comments are listed as following:

Comment 1
In section 2 (theoretical background), section 2.1 I think can be improved considerably. On the one hand, the authors speak in several paragraphs of the crossed alliance in terms of definition (lines 107-109 and lines 127-129). I believe that this information should be better reorganized.  

Authors’ response

Thank you for your careful reading of our manuscript. In the revised manuscript, we have rewritten the section 2.1, especially in the definition of brand association, the definition of fashion brand crossover alliances. For your convenience, we have included the section below:

2.1. Crossover Alliance of Fashion Brands

    Brand association originates from the concept of symbiotic marketing [9], which refers to an effective way for two or more brands to form a short-term or long-term collaboration with the aim of creating synergies and complementing each other's resources to broaden the market and increase the brand value [3,25-27]. The number of business practices and academic research literature on brand alliances has increased significantly over the past few years, as companies have become increasingly aware of the growth opportunities offered by brand alliance strategies [28]. Previous studies have focused on exploring the impact of factors such as consumers' brand loyalty, perceived brand quality, and brand fit on brand alliance success [5,29]. It is emphasized in the study that among brand association partners, either brand image fit, product category fit, or consumer perception fit affects consumer perception fluency and further influences brand association attitudes [8,30]. However, with the evolution of the business market, high fit brand alliances in the field of fashion marketing are no longer sufficient to meet market and consumer needs. For instance, Moon and Sprott (2016) [31] showed that luxury brands may find it almost impossible to collaborate with other luxury brands and therefore need to consider non-luxury brand partners. Co-partners with too similar brand images or product types may struggle to inspire freshness in consumers and thus reduce purchase intentions [32]. Further, it has been emphasized that moderate dissonance has a positive and interesting value. Studies Srivastava and Sharma (2012) [33] and Walchli (2007) [34] pointed out that in the brand extension process, the degree of brand extension consistency shows an inverted U-shaped relationship with brand extension evaluation, i.e., moderate incongruence leads to better brand evaluation than homogeneity or extreme incongruence. Based on product innovation domain theory, Zhu and Wang (2016) [35] emphasizes the importance of consumer novelty perceptions in the brand innovation process. Consumers may have stronger novelty perceptions and hedonic perceptions for innovative products that deviate from the norms. These phenomena may be due to the possibility that consumers find the brand or product more unique, iconic, or stylish [36,37].

    As a result, from luxury brands to fast fashion brands, all companies try to break the original brand image, seek differentiation, and create a "brand crossover alliance" [38]. Brand crossover alliance is a form of brand alliance, which is defined by scholars as the joint development of new products by two or more brands in non-competing industries that highly integrate the characteristics or attributes of the joint brand [39]. Further, some scholars argue that "crossover" cannot be simply summarized as the difference between industries or sectors in an objective sense. The so-called crossover should be a brand association partner that consumers perceive to be less related, similar, compatible, or complementary in terms of their salient attributes [40]. In this way, a crossover alliance between fashion brands can be a collaboration between two brands with widely different images in the fashion industry. For example, Gucci and The North Face are both fashion brands, but in terms of brand image, one is a luxury brand and the other is an outdoor sports brand. The perceived similarity of their brand images is rather small, and the product positioning or product values are very different. Oeppen and Jamal (2014) [41] and Mrad et al. (2019) [42] explored through qualitative research the good results achieved by fast fashion brands and luxury brands through low-scale collaborations. This is because it seems that through short-term collaborations luxury brands can be protected from being viewed negatively by their existing consumer base, while also attracting potential consumers to new markets through the mass market of fast fashion brands. On the other hand, the crossover alliance of fashion brands can also be the collaboration of the fashion industry with other industries. For example, Balenciaga and Moncler have successively designed virtual clothing with the online game Fortnite. In terms of product type, one is an Internet game company and the other is a luxury fashion brand, and both product types have strong differences in terms of product attributes and product usage scenarios. The case study by Moon and Sprott (2016) [31] shows that when fashion brands and brands from other sectors unite, consumers' perceived brand image fit positively influences purchase intentions for co-branded new products. The study by Paydas (2021) [21] shows that brand alliances of different product types positively influence consumers' perceptions of product quality and enhance willingness to purchase the product.

    Reviewing the above literature, we find that scholars analyze the effectiveness of fashion brand crossover alliances from the perspective of brand image difference or product type difference, respectively, but fewer scholars incorporate the two dimensions into one model for exploration and comparison. Thus, this study explores fashion brand crossover alliances from two dimensions: brand image difference and product type difference, and defines fashion brand crossover alliances as the joint development of new products by a fashion brand with single or multiple brands that have outstanding differences at the brand image level or product type level. Differences at the brand image level refer to consumers' perceptions that the brand image, brand positioning and brand perception of the partners are not similar; differences at the product type level refer to consumers' perceptions that the product attributes and product usage contexts of the partners are not compatible and complementary.

Comment 2

On the other hand, they end section 2.1 by talking about the differences in brand image and type of product, which I consider to be a very important part and therefore could be explored further. According to the authors, there are two dimensions (line 126). What authors or previous studies support these dimensions? Is there specific literature on these dimensions? I have no comment to make on the rest of the sections. 

Authors’ response

According to reviewer’s comment, we have added some previous studies as support for these dimensions. For your convenience, we have included the section below:

    In this way, a crossover alliance between fashion brands can be a collaboration between two brands with widely different images in the fashion industry. For example, Gucci and The North Face are both fashion brands, but in terms of brand image, one is a luxury brand and the other is an outdoor sports brand. The perceived similarity of their brand images is rather small, and the product positioning or product values are very different. Oeppen and Jamal (2014) [41] and Mrad et al. (2019) [42] explored through qualitative research the good results achieved by fast fashion brands and luxury brands through low-scale collaborations. This is because it seems that through short-term collaborations luxury brands can be protected from being viewed negatively by their existing consumer base, while also attracting potential consumers to new markets through the mass market of fast fashion brands. On the other hand, the crossover alliance of fashion brands can also be the collaboration of the fashion industry with other industries. For example, Balenciaga and Moncler have successively designed virtual clothing with the online game Fortnite. In terms of product type, one is an Internet game company and the other is a luxury fashion brand, and both product types have strong differences in terms of product attributes and product usage scenarios. The case study by Moon and Sprott (2016) [31] shows that when fashion brands and brands from other sectors unite, consumers' perceived brand image fit positively influences purchase intentions for co-branded new products. The study by Paydas (2021) [21] shows that brand alliances of different product types positively influence consumers' perceptions of product quality and enhance willingness to purchase the product.

Comment 3

In section 4 (methodology), the authors talk about “questions for each variable adapted from the literature”, what adaptations were necessary? What process was used for adaptation? This tool or any of its latent variables, in what previous studies has it been used? With what psychometric results? It would be interesting to include information on this. 

Authors’ response

    The reason for the adaptation of the research questions is mainly because the background of our research is not quite consistent with the previous research.

    The content of most of our questions is based on the content of previous research questions, and only adapted the scenario. For instance, the previous research on brand crossover alliance was based on the background of FMCG products, and we started our research based on clothing and fashion products. On this occasion, we exclusively altered the products and made slight changes.

    In the initial questionnaire preparation stage, we invited experts and PhD students in the field of marketing to make semantic corrections to the questionnaire. Meanwhile, we distributed 50 copies of the pre-experimental questionnaire before distributing the official questionnaire to ensure the rigor and feasibility of the questions.

Comment 4
Regarding the most well-known alliance cases, is there any specific criteria for using those cases and not others? 

Authors’ response

    The cases we chose are relatively new cases of corporate practice, selected in discussion with experts in the apparel field and PhD students.

    We added the cases to facilitate consumers' understanding of the concept of cross-border association of fashion brands, but there may be some emotional factors such as consumer familiarity with the brands that interfere with the research process. Therefore, we have included this issue in the limitations of the study, and we will consider it more in future studies. Thank you very much for your suggestion. For reviewer's convenience, we have included the section below:

7.Limitation and Future Expectations

    First, this study was conducted by means of a web-based questionnaire. The study selected six cases of crossover alliances of fashion brands that have occurred in recent years and are more typical and well-known, in order for consumers to understand the concept of crossover alliances of fashion brands. In the research process, however, some emotional factors such as consumers' familiarity with the brands may interfere. In future studies, longitudinal data and experimental methods can therefore be used to verify the conclusions of this paper.

Comment 5
Regarding the screening question, how many people indicated “no”? (identify how many people indicated “no” and how many people gave incomplete answers). 

Authors’ response

Following this suggestion, the sentence “a valid sample of 526 respondents was obtained by excluding 74 respondents who did not match or gave incomplete answers.” has been corrected as “After eliminating 21 non-conforming questions and 53 questionnaires with incomplete responses, a valid sample of 526 people was finally obtained.”

Comment 6
In section 4.2, is it possible to include the CFA adjustment information as well? Use the same indices as for the SEM analysis in section 4.3.

Authors’ response

Following this suggestion, we have added the model fit indices for CFA analysis in Section 4.2. For reviewer's convenience, we have included the section below:

    First, the factor measurement model had a good fit (χ2=241.467; χ2/df=1.700; p=0.000; GFI=0.955; AGFI=0.940; CFI=0.991; TLI=0.989; RMSEA=0.037), and therefore, a reliability test could be performed.

Comment 7
The limitations do not indicate anything about the sample and its composition. Is there any limitation that 75% of the sample is between the ages of 18 and 39?

Authors’ response

There is no limitation on the age of 18-29. We just made a simple statistical description.

Comment 8
The references do not conform to the journal's regulations. Please review this section completely.

Authors’ response

Thank you for your careful reading of our manuscript. We have revised the format of all references.

Once again, thank you very much for your valuable comments and detailed suggestions. And we hope our correction will meet the approval.  

References

  1. Ahn, J.; Kim, A.; Sung, Y. The effects of sensory fit on consumer evaluations of co-branding. j. advert. 2019, 39, 486-503, doi:10.1080/02650487.2019.1652518.
  2. Paydas Turan, C. Success drivers of co-branding: A meta-analysis. Int. j. consum. stud. 2021, 45, 911-936, doi:10.1111/ijcs.12682.
  3. Ma, B.; Cheng, F.; Bu, J.; Jiang, J. Effects of brand alliance on brand equity. Journal of Contemporary Marketing Science 2018, 1, 22-33, doi:10.1108/JCMARS-08-2018-0007.
  4. Adler, L. Symbiotic marketing. Harvard. bus. rev. 1966, 44, 59-71.
  5. Paydas Turan, C. What's inside matters: The impact of ingredient branding on consumers' purchasing behaviours in services. J. retail. consum. serv. 2021, 63, 102690, doi:10.1016/j.jretconser.2021.102690.
  6. Rao, A.R.; Ruekert, R.W. Brand alliances as signals of product quality. Mit. sloan. manage. rev. 1994, 36, 87-97.
  7. Rao, A.R.; Qu, L.; Ruekert, R.W. Signaling Unobservable Product Quality through a Brand Ally. J. marketing. res. 1999, 36, 258-268, doi:10.1177/002224379903600209.
  8. Newmeyer, C.E.; Venkatesh, R.; Ruth, J.A.; Chatterjee, R. A typology of brand alliances and consumer awareness of brand alliance integration. Market. lett. 2018, 29, 275-289, doi:10.1007/s11002-018-9467-4.
  9. Newmeyer, C.E.; Venkatesh, R.; Chatterjee, R. Cobranding arrangements and partner selection: a conceptual framework and managerial guidelines. J. acad. market. sci. 2014, 42, 103-118, doi:10.1007/s11747-013-0343-8.
  10. Eren-Erdogmus, I.; Akgun, I.; Arda, E. Drivers of successful luxury fashion brand extensions: cases of complement and transfer extensions. J. fash. mark. manag. 2018, 22, 476-493, doi:10.1108/JFMM-02-2018-0020.
  11. Ahn, J.; Kim, A.; Sung, Y. The effects of sensory fit on consumer evaluations of co-branding. Int. j. advert. 2020, 39, 486-503, doi:10.1080/02650487.2019.1652518.
  12. Moon, H.; Sprott, D.E. Ingredient branding for a luxury brand: The role of brand and product fit. J. bus. res. 2016, 69, 5768-5774, doi:10.1016/j.jbusres.2016.04.173.
  13. Sénéchal, S.; Georges, L.; Pernin, J.L. Alliances Between Corporate and Fair Trade Brands: Examining the Antecedents of Overall Evaluation of the Co-branded Product. J. bus. ethics. 2014, 124, 365-381, doi:10.1007/s10551-013-1875-z.
  14. Srivastava, K.; Sharma, N.K. Consumer attitude towards brand-extension incongruity: The moderating role of need for cognition and need for change. J. market. manag-uk. 2012, 28, 652-675, doi:10.1080/0267257X.2011.558383.
  15. Walchli, S.B. The effects of between-partner congruity on consumer evaluation of co-branded products. P&M. 2007, 24, 947-973, doi:10.1002/mar.20191.
  16. Zhu, Q.; Wang, X.-y. The Influential mechanism of perceived product Innovativeness on consumers’purchase intention: The moderating role of country-of-origin image and price sensitivity. Business and Management Journal 2016, 38, 107-108, doi:10.19616/j.cnki.bmj.2016.07.010.
  17. Warren, C.; Batra, R.; Loureiro, S.M.C.; Bagozzi, R.P. Brand Coolness. J. marketing. 2019, 83, 36-56, doi:10.1177/0022242919857698.
  18. De Veirman, M.; Cauberghe, V.; Hudders, L. Marketing through Instagram influencers: the impact of number of followers and product divergence on brand attitude. Int. j. advert. 2017, 36, 798-828, doi:10.1080/02650487.2017.1348035.
  19. Sharma, A.; Soni, M.; Borah, S.B.; Saboo, A.R. Identifying the drivers of luxury brand sales in emerging markets: An exploratory study. J. bus. res. 2020, 111, 25-40, doi:10.1016/j.jbusres.2020.02.009.
  20. Zhou, Y.; Zhang, Y.; Chen, S. Influence of product association distance on the evaluation of joint product and consumers' purchase intention in the context of brand crossover alliance. Journal of Business Economics 2021, 37-50, doi:10. 14134 /j. cnki. cn33-1336 /f. 2021. 12. 003.
  21. Jian, Y.; Zhu, L.; Zhou, Z. The mechanism of brand alliance attitude towards Cross-boundary Co-development: Based on the Perspectiveof Consumer Inspiration Theory. Nankai. bus. rev. nt. 2021, 24, 25-38, doi:10.3969/j.issn.1008-3448.2021.02.004.
  22. Oeppen, J.; Jamal, A. Collaborating for success: managerial perspectives on co-branding strategies in the fashion industry. J. market. manag-uk. 2014, 30, 925-948, doi:10.1080/0267257X.2014.934905.
  23. Mrad, M.; Farah, M.F.; Haddad, S. From Karl Lagerfeld to Erdem: a series of collaborations between designer luxury brands and fast-fashion brands. J. brand. manag. 2019, 26, 567-582, doi:10.1057/s41262-018-00146-2.

Round 2

Reviewer 2 Report

I keep my initial review: The manuscript does not seem to be genuine and the content is not credible (it does not seem to be based on an actually conducted research). It only pretends to be scientifically sound without convincing of actually being. It mimics scientific conventions without providing valuable content for the field. 

Author Response

RESPONSES TO REVIEWER # 2

Dear Reviewer,

        We have studied the valuable comments from you carefully, and tried our best to revise the manuscript, which we hope meet the approval.

Comment

        The manuscript does not seem to be genuine and the content is not credible (it does not seem to be based on an actually conducted research). It only pretends to be scientifically sound without convincing of actually being. It mimics scientific conventions without providing valuable content for the field.

Authors’ response

        In recent years, there are frequent successful cases of crossover alliances based on brand level or product level in the brand practice of fashion industry. In terms of brand image difference, for example, the case of cooperation between fast fashion and designer luxury brands has gained much media attention [1]. UNIQLO had a two-year-long cooperation with German luxury designer brand Jil Sander. Taking their co-brand +J as an example, UNIQLO, after cooperating with Jil Sander, should strengthen its uniqueness through marketing and brand promotion, thus attracting consumers to purchase +J’s productions repeatedly [2]. The fashion industry is much more extensive in terms of product type differences, such as the collaboration between German sports brand Puma and Japanese game company Nintendo to launch a co-branded sneaker line. Dior and Italian motorcycle brand Piaggio also jointly launched a vespa motorcycle with the Dior logo. These real-life examples show that brands can also achieve good results by choosing alliance partners that are different in terms of both product and brand image.

        Previous studies on brands’ crossover alliances are more often found in the food and electronics industries. For instance, Jian [3] studied both brand image and product type, which demonstrated that many brand alliances in the food industry with differences in brand image or product type resulted in positive consumer attitudes toward the brand. However, in general, previous studies have focused more on consumers' attitudes towards crossover alliances or their willingness to buy.

        This study focuses on fashion brands, which are less consistent with food or electronics categories, and consumer demand for their attributes. Fashion products are hedonic products, and fashion design helps distinguish itself from others. It expresses abstract concepts of self-taste, identity, temperament, and aesthetic tendencies from what people wear. In the meanwhile, young people consume fashion not only because of the product itself, but for the symbolic meaning of specific clothing [4,5].

        Interestingly, comparing with profit, in the fashion industry the attention of social media and mass media gained by brand alliance is more crucial to measure the success of joint projects [6]. Social media provides an opportunity for brands to enhance their brand value by offering an online platform to share information and exchange ideas among users, regardless of time, place and medium [7]. Social media has been ranked as the number one influence lever to affect Chinese consumers' opinions and decisions on fashion product purchases [8]. Consumers' brand engagement on social media, to some extent, helps brands to respond to emerging market trends and reposition their brand image [9]. In this way, they can attract new and diverse consumer groups [10]. Therefore, based on the context of crossover alliances of fashion brands, we believe that it is more theoretical and practical to study consumers’ online brand engagement.

        Based on previous studies, this study further explores the extended dependent variables of consumer brand attitudes, and purchase intentions consumer online brand engagement in the context of fashion brand crossover alliances. Using stimulus-organism-response (S-O-R) theory, two mediating variables, consumer novelty perception and hedonic perception, are introduced in this study. The study purpose is to explore the potential mechanisms by which brand image differences and product type differences in fashion brand crossover alliance affect the formation of consumer online brand engagement. On the one hand, this study enriches the relevant research in the field of crossover alliance from the perspective of fashion brands. On the other hand, it further explores the mechanisms of fashion brand crossover alliance on consumers' online brand engagement. By doing so, it expands the application of fashion brand crossover alliance in the field of consumer online brand engagement research.

        Once again, thank you very much for your valuable comments and detailed suggestions. And we hope our correction will meet the approval.  

References

  1. Lee, C.-L.; Decker, R. CO-BRANDING PARTNER SELECTION: THE IMPORTANCE OF BELIEF REVISION. Journal of Business Economics and Management 2016, 17, 546-563, doi:10.3846/16111699.2016.1197848.
  2. Shen, B.; Jung, J.; Chow, P.-S.; Wong, S. Co-branding in Fast Fashion: The Impact of Consumers’ Need for Uniqueness on Purchase Perception. In Fashion Branding and Consumer Behaviors: Scientific Models, Choi, T.-M., Ed.; Springer New York: New York, NY, 2014; pp. 101-112.
  3. Jian, Y.Z., LiyaZhou,Zhiming. The mechanism of brand alliance attitude towards Cross-boundary Co-development: Based on the Perspectiveof Consumer Inspiration Theory. Nankai Business Review 2021, 24, 25-38, doi:10.3969/j.issn.1008-3448.2021.02.004.
  4. Khallouli, K.J.; Gharbi, A. Symbolic consumption by teenagers: A discussion through the optics of appearance and identity. International Journal of Business and Social Science 2013, 4.
  5. Roux, D. Am I what I wear? An exploratory study of symbolic meanings associated with secondhand clothing. ACR North American Advances 2006.
  6. Mrad, M.; Farah, M.F.; Haddad, S. From Karl Lagerfeld to Erdem: a series of collaborations between designer luxury brands and fast-fashion brands. Journal of Brand Management 2019, 26, 567-582, doi:10.1057/s41262-018-00146-2.
  7. Zhang, L.; Zhao, H.; Cude, B. Luxury brands join hands: building interactive alliances on social media. Journal of Research in Interactive Marketing 2021, 15, 787-803, doi:10.1108/JRIM-02-2020-0041.
  8. Bianchi, N.P.A.W.J.S.O.A.F. True luxury global consumer insights 2018. Available online: https://media-publications.bcg.com/france/True-Luxury-Global-Consumer-Insight-2018.pdf (accessed on 20 February ).
  9. Voss, K.E.; Mohan, M. Corporate brand effects in brand alliances. Journal of Business Research 2016, 69, 4177-4184, doi:10.1016/j.jbusres.2016.03.007.
  10. Yu, Y.; Rothenberg, L.; Moore, M. Exploring young consumer's decision‐making for luxury co-branding combinations. International Journal of Retail & Distribution Management 2021, 49, 341-358, doi:10.1108/IJRDM-12-2019-0399.

Round 3

Reviewer 2 Report

The authors not improve the research on which the manuscript is based (the paper is not credible as being based on an actual research - it is just an imitation of the following paper: https://www.mdpi.com/2071-1050/11/20/5830)